# Fluctuations in National Resilience during the COVID-19 Pandemic

**DOI:** 10.3390/ijerph18083876

**Published:** 2021-04-07

**Authors:** Shaul Kimhi, Yohanan Eshel, Hadas Marciano, Bruria Adini

**Affiliations:** 1Stress and Resilience Research Center, Tel-Hai College, Northern Galilee 122800, Israel; yeshel@psy.haifa.ac.il (Y.E.); hmarcia1@univ.haifa.ac.il (H.M.); 2Department of Psychology, University of Haifa, Haifa 3498838, Israel; 3Ergonomics and Human Factors Unit, University of Haifa, Haifa 3498838, Israel; 4Department of Emergency and Disaster Management, School of Public Health, Sackler Faculty of Medicine, Tel Aviv University, Tel Aviv 6139001, Israel; adini@netvision.net.il

**Keywords:** national resilience, COVID-19, political, economic, health threat

## Abstract

The current study measured national resilience (NR) in three different time frames during the coronavirus disease 2019 (COVID-19) pandemic in Israel (*N* = 804). We investigated two main issues: first, the direction and extent of NR changes during the crisis, and second, the predictors of NR. The results show the following: (a) the average NR score declined significantly across the three repeated measures, with a medium-size effect. (b) Three of the four identified NR factors declined significantly across the three measurements: belief in the government and the prime minister (large effect size); belief in civil society; and patriotism (medium effect size); while trust in Israeli national institutions was the lowest and did not weaken significantly. (c) Analyzing the prediction of NR factors indicated that the levels of the three NR factors mainly reflected one’s political attitudes, sense of political and economic threats, rather than health threats. One conclusion concerns the importance of trust in leadership as the most sensitive component in the decline of national resilience following a crisis.

## 1. Introduction

As of writing (March 2021), it has been more than a year since the coronavirus pandemic was first identified in Wuhan, China, and then worldwide. The pandemic became an unprecedented threat in its scope and degree of damage, affecting many areas of our routine lives [1].

The coronavirus disease 2019 (COVID-19) outbreak in Israel began on 27 February; as of 25 March 2021, there were 6163 dead as a result of the pandemic. Israel was ranked globally (as of 30 November 2020) 48th in the mortality prevalence from COVID-19 (311 people per million inhabitants died), out of more than 200 countries affected by this virus. We have examined distress and resilience using three repeated measurements conducted during the COVID-19 pandemic: in May 2020, July 2020, and October 2020.

The focus of the research was to investigate fluctuations in the national resilience of civil society as well as to determine whether it was affected mainly by the health crisis, or by other concurrent crises, such as political or economic. All these crises signify some of the most severe crises experienced by Israel [2]. Following the global epidemic, much attention was paid to the issue of resilience.

### 1.1. Resilience

Resilience is a theoretical concept that allows us to explain and even predict variability between human beings in their ability to deal with adversities. A literature review indicates that there are many definitions of resilience [3]. The professional literature often distinguishes between three levels of resilience: Individual resilience refers to the individual’s ability to successfully cope with adversity and return to full function after the event. Individual resilience is influenced by many biological and environmental factors and is the area most researched amongst the resilience realm [4]. Community resilience refers to the way a community manages damage and disruption that impacts on populations, buildings, infrastructure, economy, societal facilities and services, in a geographical area that includes local leadership and a mechanism to effectively respond to the adversity [5,6]. Throughout the past decade, the definition of ‘community resilience’ has expanded from the ability of the community to “prepare for and adapt to changing conditions and to withstand and recover rapidly from disruptions” to also include mitigation of the impacts or consequences, as well as reduce future vulnerability [6]. National resilience (NR) refers to the ability of an entire country to deal with adversity and recover as quickly as possible. However, NR is the level of resilience that has been least studied [7]. National resilience has been recognized as a complex phenomenon, that is founded on coping abilities and cohesion of varied elements of society [8]. Canetti, Waismel-Manor, Cohen, and Rapaport [9] suggested that national resilience represents a country’s capacity to effectively manage different types of crises (for example human-made conflicts, poor socio-economic conditions, or negligence), simultaneously with sustaining the function of society as a whole. Another definition of the concept of NR relates to the need to adapt to and modify the societal function while balancing expectations, capacities, and rising challenges [10]. A crucial component of the NR is the level of trust of the population in its national leadership and institutions, as well as in the bodies of civil society itself [11,12].

Since the outbreak of the coronavirus crisis, the concept of national resilience has received much attention, in the aim of measuring a country’s ability to deal with the pandemic that constitutes a multidimensional crisis [13].

### 1.2. Predictors of National Resilience

In an earlier study, we examined the demographic and psychological predictors of national resilience. Levels of religious and political belief, as well as socio-economic status, were found to be demographic predictors of NR. Perceived coherence was found to be a psychological predictor of both community and national resilience [14]. These findings were also presented by Marciano, Eshel, and Kimhi [15] who also found two additional attributes, including concern about conflicts and experience of terrorism.

The present study focuses on national resilience throughout the COVID-19 pandemic, to highlight changes in this domain throughout the prolonged crisis. Specifically, we examined the following main questions: Has the overall NR and its components declined continuously as the pandemic continued? Did they change throughout the pandemic to similar extents? What variables predicted the varied components? To the best of our knowledge, these issues have not been explored so far.

This research examined national resilience during “multidimensional crises”: health, economic, and political [16]. More specifically, we focused on the question of which of these hardships had a greater impact on changes that occurred to the Israeli NR amidst the ongoing COVID-19 pandemic.

Four potentially relevant predictors were employed in the current study: a current sense of health threat, a recent sense of economic threat, a present sense of political threat, and declared political attitudes. A previous study, based on two repeated measurements, indicated an ongoing trend of a decrease in Israeli NR during the pandemic [17]. However, the previous study was conducted two months after the peak of the crisis, upon the initial phase of lifting the lockdown, while the current study takes place toward the end of the second lock-down. Very few studies have been conducted to date that enable us to understand the mechanisms of national resilience over time, in a prolonged crisis. Therefore, the third measurement allows us to examine the long-term impact of the coronavirus crisis concerning national resilience and to what extent the varied threats (health, economy, and political crisis) predict the national resilience: If the decrease of NR reflects people’s sense of inability to overcome the COVID-19 pandemic (i.e., fear of the virus and its potential consequences), we would expect that the health threat will be the best predictor of this decrease. However, our previous results indicated that the decrease in NR is best predicted by perceived political and economic factors. Thus, it is important to investigate whether this phenomenon is a singular occurrence or whether it actually represents a more lasting manifestation.

### 1.3. Hypotheses and Research Question

Based on the existing literature, the following hypotheses were examined:National resilience would decrease significantly throughout the three repeated measurements during the COVID-19 pandemic.Each national resilience factor would decline from the first to the third measurement.

An additional research question would ask which threat would better predict the four NR factors: a health threat or a more politically oriented threat?

## 2. Materials and Methods

### 2.1. Study Design

The current research explored paired (identical) samples of participants from Israel that filled the survey during three-time frames during the COVID-19 pandemic. The initial measurement (T1) was conducted two months after the onset of the pandemic (4–7 May), upon the decrease in the social distancing measures, including the exit from the national closure that was previously implemented. The subsequent assessment (T2) was measured during the “second wave” of the pandemic that was characterized by a rise in levels of infectivity and confirmed COVID-19 cases (12–15 July), which resulted in renewed social distancing measures, on the overall nation’s population. The third measurement (T3) was conducted in midst of the second lockdown (12–14 October).

### 2.2. Participants

The data were assimilated by an internet panel corporation that consists of more than 65,000 members, from the varied groups of the Israeli population (https://sekernet.co.il/ accessed on 4 April 2021). The sample included Jewish Israeli respondents, who responded three times to an online questionnaire. Before its distribution, the study was approved by the Tel Aviv University’s Ethics Board.

### 2.3. Study Tools

#### 2.3.1. National Resilience

The index used for measuring national resilience included 16 parameters (NR-16), that were previously validated [18]. The parameters range from 1 = do not agree at all to 6 = strongly agree. The index’s internal reliability was high, in all three-time frames in which it was used (α = 0.91). A factor analysis was conducted on the 16 items of NR based on the findings of T1 [11].

#### 2.3.2. Demographic Characteristics

Nine demographic attributes were collected: age (categorized to 4 age groups), gender, level of religiosity (1 = non-religious to 4 = highly religious), socio-economic level compared to the mean level (1 = much lower than the national average to 5 = much higher than the national average), political attitudes (1 = extreme left to 5 = extreme right), level of education (1 = elementary to 5 = graduate degree and higher), familial status (single, married, divorced, in a relationship), number of children (no children to 4 children or more), and economic difficulties due to the COVID-19 pandemic (1 = not at all, to 5 = to a very much degree). Table 1 describes the attributes among the participating respondents.

### 2.4. Data Analysis

We used three statistical tests to investigate the results as per each hypothesis: (a) General linear repeated model across three repeated measures of NR-16 average items. (b) Distribution and general linear model of three repeated measurements of average overall national resilience. (c) Distribution of three repeated measurements of R factors. (d) Path analysis of political attitudes and three threats at T3 predicting NR factors at T3.

## 3. Results

The survey included 804 respondents that completed the same questionnaire in all three measurements. The description of the sample characteristics is described in Table 1.

Factor analysis on the 16 items of NR at T1 (principal component and Varimax rotation) revealed four factors as follows (Table 2): Factor 1 ‘belief in the government and the prime minister’ (8 items, explaining 25% of NR variance, Alpha Cronbach 0.88 to 0.89); factor 2 ‘belief in the civil society’ (4 items, explaining 17% of NR variance, Alpha Cronbach 0.80 to 0.87); factor 3 ‘patriotism’ (3 items, explaining 15% of NR variance, Alpha Cronbach 0.80 to 0.82), and factor 4 ‘belief in public Israeli institutions’ (3 items, explaining 11% of NR variance, Alpha Cronbach 0.60 to 0.88). The factor analysis revealed the following: (a) Factors 1, 2, and, 4 showed near-normal distributions, while factor 3 (patriotism) showed negative skew distribution (most participants reported a very high level of patriotism). (b) The correlations between the four factors indicate that factor four correlated lower with the three others, compared with the intercorrelations among the others. (c) The four factors explained 69% of NR variability. (d) The factor loading across the three repeated measurements was similar.

### Predictors of National Resilience (NR) Factors

Three of the four predicting items of NR were phrased as follows: “To what extent would you rate today the current economic condition as threatening you personally?”; “To what extent would you rate today the current health condition as threatening you personally?”; “To what extent would you rate today the current political condition as threatening you personally?”. These items were rated by a 5-point Likert- scale in which 1 = “Not threatening at all”, and 5 = “Threatening very much”. The fourth predictor was phrased: “How would you rate yourself politically, as far as security and external issues are concerned?” The 5-point Likert scale for this item ranged from 1 = “Extreme left” to 5 = “Extreme right”.

Next, we calculated the averages and standard deviation of all NR-16 items, across the three repeated measures (Table 3). The results indicated the following: (a) The three items that decreased mostly over the three measurements pertained to trust in the political leadership. (b) The three items of national resilience with the highest mean level related to belief in the Israeli society (patriotism), while the three lowest NR items that dealt with belief in the Israeli national institutions, especially the parliament. It was found further that the item ‘trust in the Israeli parliament (Knesset)’ scored lowest across the three measurements, whereas the item that scored highest was ‘Israel is my home, and I do not plan to leave it’.

To investigate the first hypothesis, the General Linear Model was calculated for the three assessments (Table 4). The findings were as follows: (a) The average NR score decreased significantly throughout the three repeated measurements: (b) Post hoc Scheffe analysis indicated significant differences between the three measurements (*p* < 0.001). (c) A medium-size decrease effect was found (η_p_^2^ = 0.38). (d) The average score in all three measurements was above 3 (the midpoint on the scale 1–6). In other words, the high level of NR in the first measurement decreased to a medium level at the third measurement. (e) The average national resilience score tended to be normally distributed across the three measurements.

We also examined to what degree each of the four factors of NR fluctuated across the varied assessments (Table 5). The results indicated the following: (a) Factor 1, (belief in the government and the prime minister), factor 2 (belief in the civil society), and factor 3 (patriotism) declined significantly across these three repeated measurements. (b) The decline of factor 1 showed the largest effect size, factor 3 showed a medium effect size, while factor 2 had the smallest effect size. (c) Factor 4 (trust in Israeli national institutions) did not change significantly across the three measurements, and it was always below the medium of the scale (less than 3). (d) A further examination of the level of national resilience among each of the average factors indicates that the mean score of the patriotism factor was higher across all the three repeated measures, compared with the other factors, while trust in national institutions in Israel was the lowest.

The question of whether the recent decline of national resilience in Israel reflected mainly the current political, economic, or health (pandemic) crisis, was examined by a path analysis [19]. In this analysis political attitudes and perceived economic, health, and political threats predicted the four NR factors. The results (Table 6) indicated the following: (a) The best predictor of factor 1 (belief in the government and the Prime Minister) was the political attitude: the more rightwing the attitudes, the higher level of trust reported. The second-best predictor of factor 1 was the political threat. A higher level of trust in the state and its leader was predicted by a lower perceived political threat. The four predictors explained 21% of factor 1 variability. (b) These results were replicated in predicting factor 2 (trust in the Israeli society). As displayed concerning factor 1, the best predictor of factor 2 was also political attitudes: the more rightwing the attitudes, the higher level of trust in Israeli society. The second-best predictor of factor 2 was the political threat. A higher level of trust in Israeli society was predicted by a lower perceived political threat. The four predictors explained 15% of factor 2 variability. (c) As found for the first two factors, the best predictor of factor 3 (patriotism) was political attitudes: the stronger the rightwing perceptions, the higher the declared patriotism. The second-best predictor of this factor was the economic threat: the lower the level of perceived economic threat due to the COVID-19 crisis, the higher was patriotism. The four-predictors explained 17% of factor 3 variability. (d) The best predictor of factor 4 (trust in Israeli national institutions) was the economic threat: the higher the perceived economic threat, the lower the trust in Israeli national institutions. The second-best predictor of this factor was the health threat: the higher the perceived health threat, the higher the level of trust in Israeli national institutions. The four-predictors explained only 0.02% of factor 4 variability.

## 4. Discussion

The current research investigated fluctuations in the level of national resilience throughout three assessments, along with the COVID-19 crisis, among a large cohort of Israeli civil society. The results showed a steady and significant reduction in the average levels of national resilience during the prolonged pandemic.

Following earlier studies that examined national resilience [20] or social resilience [21], our study indicated a complex structure of national resilience that includes various distinct components or factors. Thus, it was found that the decrease in the level of resilience of each of the four factors differed across the three repeated measures, ranging from a high decrease (for the factor ‘trust in the state and its leader’) to no change (for the factor ‘trust in Israeli national institutions’).

Aligned with previous studies, the present study indicates that the decline in belief in the government and the prime minister was the most prominent among the four factors, and points to the great importance of trust in leadership during severe adversities, such as the COVID-19 crisis [22]. Our research indicates that, during the prolonged pandemic, the level of belief in leadership is the most sensitive component of national resilience in Israel.

At the same time, trust in Israeli national institutions was found to be the lowest across the three measurements, showing no significant change across the various measurement times. This finding may be attributed to the prolonged instability in the Israeli political arena, while the public and national institutions are perceived as investing efforts in maintaining their functional continuity as much as possible, including between and during lockdowns [23]. However, further longitudinal studies in other cultures are needed to support these findings.

National ability to effectively manage crises, while keeping the functionality of civil society, was often challenged by several simultaneous countrywide emergencies rather than by a single hardship [24]. Israel is currently facing three concurrent major adversities: the COVID-19 pandemic, and a political as well as an economic crisis. Previous research indicated that efforts to cope with COVID-19 in Israel were judged contingent on political attitudes [16]. However, determining which of the three national adversities constitutes the main attribute for the decreased NR scores, remained an open question [25]. The present study addressed this issue by examining whether the four NR factors were predicted primarily by health considerations, as expected by several authors [26], or by political and economic concerns [27]. Our data indicated that in the present case, national resilience was enhanced by supporting the political perspective of the Israeli government, and by believing that Israel currently faces neither a political nor an economic crisis. Thus, our data suggest that political concerns mostly affected the NR, and that despite the COVID-19 pandemic, the four investigated NR factors were more marginally affected by a sense of health threats. In other words, the ongoing political conflict in Israel was a major contributor to the decline of its NR level rather than its failure to cope consistently with the ongoing pandemic or its repercussions on the economic system. This conclusion concerning the effects of the COVID-19, which reflects the impact of concurrent national adversities, should be examined in other countries to establish its generalizability.

## 5. Limitation

Four limitations of this research should be noted: (1) A correlative study does not allow inference about causality. (2) The study was based on an internet panel’s sample rather than on a random sample. (3) The lack of publications concerning other longitudinal research initiatives in other countries, does not enable multi-country comparisons, that would have contributed towards generalization of the results, identification of varied preferences, and cultural contextures. (4) The Cronbach Alpha of the subscale ‘belief in public Israeli bodies’ was 0.60, constituting a moderate level; most probably this is derived from the scale being based on only three items.

## 6. Conclusions

The present study draws several main possible conclusions: the first refers to the complexity of national resilience as a psychological structure such as trust in the state leadership. The study examined the varied dimensions or factors of national resilience beyond the overall score. The second refers to the fact that an ongoing and difficult crisis may involve a decline in national resilience, as suggested by the current study. The continual decrease in national resilience strengthens the need to conduct ongoing studies to examine national resilience throughout crises. The third conclusion concerns the importance of trust in leadership as the most sensitive component in the decline of national resilience following a crisis. Our findings present the element of trust in national leadership as the most important factor of national resilience. The fourth conclusion is that during adversity, it is vital to monitor whether there is only one type of crisis or several parallel ones and, accordingly, examine which crisis is perceived by the public as the most threatening. This information may be valuable both theoretically and applied. The fifth conclusion that emerges concerns the importance of longitudinal studies concerning resilience, as a single measurement indicates only part of the overall picture. Therefore, we suggest that further longitudinal studies conducted in other countries are needed to generalize the present results beyond the Israeli context.

Beyond the conclusions mentioned above, the findings of the study should be considered when designing and modifying public policies for managing varied adversities. Strengthening citizens’ trust in state leadership, regardless of the nature of the crisis, is of paramount importance in enhancing national resilience. It is vital to the effectiveness of governance systems, as without the trust of civil society in its leadership, compliance with rules and regulations are expected to be substantially hampered.

## Figures and Tables

**Table 1 ijerph-18-03876-t001:** Characteristics of the study sample (*N* = 804).

Attribute	Group	Participants	%	Mean
				(SD)
Age	1. 18–30	171	21	44.65
	2. 31–40	191	24	−15.41
	3. 41–60	151	19	
	4. 51–60	141	17	
	5. 61 on	150	19	
Gender	1. Men	416	52	
	2. Woman	388	48	
Level of Religiosity	1. Non-religious	398	49	1.81
	2. Traditional	231	29	−0.96
	3. Religious	107	13	
	4. Highly religious (orthodox)	68	9	
Political attitudes	1. Extreme left	10	1	3.49
	2. Left	87	11	−0.86
	3. Center	288	36	
	4. Right	340	42	
	5. Extreme right	79	10	
Economic difficulties due to the pandemic at T3	1. Not at all	179	22	2.61
	2. A little	209	26	−1.22
	3. Medium	236	29	
	4. Much	105	13	
	5. Very much below	75	10	
Average family income	1. Very much below	234	29	2.47
	2. Much below	180	22	−1.22
	3. Average	216	27	
	4. Much above	128	16	
	5. Very much above	46	6	

**Table 2 ijerph-18-03876-t002:** Factor analysis on national resilience (NR) comprising 16 items across the three repeated measurements.

Factor Loading/Explained Variance			
Factors, Items, and Theoretical Areas (Scale 1–6)	T1	T2	T3
Factor 1: Belief in the government and the prime minister	26%	25%	24%
1. The Israeli government and the prime minister present high leadership capacities during the coronavirus disease 2019 (COVID-19) pandemic	0.81	0.84	0.85
2. During a national crisis, such as the current coronavirus crisis, the civil society will support the decisions of the government and the prime minister	0.71	0.78	0.77
3. I have full confidence in the ability of the security forces (military) of my country to protect our population	0.57	0.45	0.47
11. I believe in the capacity of Israel’s healthcare system to provide for the medical needs of the population during the COVID-19 pandemic.	0.67	0.60	0.58
12. I fully believe in the capacity of the Israeli government to provide for all needs and succeed in containing the current COVID-19 crisis	0.78	0.82	0.82
14. Belief in the Knesset (parliament)	0.66	0.61	0.58
Factor 2: Belief in civil society	17%	17%	16%
7. Cohesiveness between the varied societal sectors is good.	0.71	0.64	0.69
8. A high level of social solidarity characterizes civil society.	0.70	0.76	0.68
9. My society is not characterized by ‘toxic relations’	0.81	0.82	0.74
10. My society is characterized by a reasonable level of social justice.	0.68	0.68	0.71
Factor 3: Patriotism	14%	14%	15%
4. Israel is my home, and I do not plan to leave it	0.84	0.82	0.85
5. Israeli society has effectively managed former crises and will manage effectively the current COVID-19 crisis	0.65	0.68	0.67
6. I am hopeful about the future of my State	0.60	0.66	0.68
Factor 4: Trust in the national institutes	11%	12%	12%
13. The police	0.75	0.77	0.68
15. The education system	0.48	0.60	0.62
16. The media	0.83	0.81	0.79
Overall national resilience scale	69%	68%	67%

**Table 3 ijerph-18-03876-t003:** Average and standard deviation of NR-16 items, across three repeated measures (arranged from high to low NR T1).

National Resilience 16 Items (Scale 1–6)	M (SD)			Change: T1–T3
	T1	T2	T3	
4. Israel is my home, and I do not plan to leave it	5.17 (1.29)	5.02 (1.43)	4.93 (1.46)	0.24
5. Israeli society has effectively managed former crises and will manage effectively the current COVID-19 crisis	4.71 (1.14)	4.00 (1.34)	3.75 (1.38)	0.96
3. I have full confidence in the ability of the security forces (military) of my country to protect our population	4.51 (1.24)	3.84 (1.46)	3.58 (1.49)	0.93
6. I am hopeful about the future of my State	4.50 (1.32)	3.97 (1.47)	3.77 (1.51)	0.73
8. A high level of social solidarity characterizes civil society.	4.19 (1.22)	3.78 (1.30)	3.45 (1.37)	0.74
11. I believe in the capacity of Israel’s healthcare system to provide for the medical needs of the population during the COVID-19 pandemic.	4.04 (1.32)	3.35 (1.40)	3.28 (1.41)	0.76
1. The Israeli government and the Prime Minister present high leadership capacities during the COVID-19 pandemic	3.96 (1.41)	2.98 (1.47)	2.76 (1.48)	1.20
12 I fully believe in the capacity of the Israeli government to provide for all needs and succeed in containing the current COVID-19 crisis	3.77 (1.49)	2.89 (1.49)	2.65 (1.50)	1.12
2. During a national crisis, such as the current coronavirus crisis, the civil society will support the decisions of the government and the prime minister	3.75 (1.27)	2.79 (1.30)	2.47 (1.33)	1.28
9. My society is not characterized by ‘toxic relations’	3.58 (1.34)	3.40 (1.34)	3.18 (1.38)	0.40
13. Trust the police	3.47 (1.46)	2.87 (1.38)	2.98 (1.48)	0.49
10. My society is characterized by a reasonable level of social justice.	3.37 (1.30)	3.09 (1.30)	2.92 (1.29)	0.45
7. Cohesiveness between the varied societal sectors are good.	3.16 (1.36)	2.81 (1.30)	2.44 (1.29)	0.72
15. Trust the education system	3.52 (1.26)	3.27 (1.28)	3.17 (1.32)	0.35
16. Trust the media	2.90 (1.41)	2.81 (1.43)	2.63 (1.44)	0.27
14. Trust in the Knesset (parliament)	2.84 1.35)	2.52 (1.32)	2.34 (1.32)	0.50

**Table 4 ijerph-18-03876-t004:** Distribution and general linear model—three repeated measurements of the average overall national resilience (*N* = 804).

National Resilience (Scale 1–6)	T1 (May 7)		T2 (July 10)		T3 (October 12)	
	Participants	%	Participants	%	Participants	%
1–2	25	3	67	8	106	13
2.1–3	125	15	229	28	252	31
3.1–4	300	37	326	40	306	38
4.1–5	300	37	161	20	125	15
5.1–6	54	7	21	8	16	2
M	3.84 ^a^		3.33 ^b^		3.14 ^c^	
SD	0.87		0.88		0.91	
Alpha Cronbach	0.88		0.89		0.90	
F_(1, 804)_ = 501.13 ***, η_p_^2^ = 0.38						

*** *p* < 0.001; ^a,b,c^ post-hoc Scheffe.

**Table 5 ijerph-18-03876-t005:** General linear model—three assessments of the national resilience factors (*N* = 804).

		T1 (May 7)	T2 (July 10)	T3 (October 12)	F_(1, 803)_	η_p_^2^
Factor 1	M	3.81 ^a^	3.06 ^b^	2.84 ^c^	882.73 ***	0.524
	SD	1.09	1.11	1.13		
Factor 2	M	3.57 ^a^	3.07 ^b^	2.99 ^c^	285.16 ***	0.262
	SD	1.05	1.06	95		
Factor 3	M	4.79 ^a^	4.33 ^b^	4.15 ^c^	378.66 ***	0.320
	SD	1.06	1.19	1.24		
Factor 4	M	2.96	2.98	2.92	1.40	0.004
	SD	0.81	1.04	1.04		

*** *p* < 0.001; ^a,b,c^ post-hoc Scheffe; Factor 1 = belief in the government and the prime minister; Factor 2 = belief in civil society; Factor 3 = patriotism; Factor 4 = belief in the national bodies.

**Table 6 ijerph-18-03876-t006:** Standardized estimates of path analyses of political attitudes and three threats at T3 predicting NR factors at T3 (present from high to low in factor 1).

Attribute	Factor 1	Factor 2	Factor 3	Factor 4
Political attitudes	0.319 ***	0.254 ***	0.329 ***	−0.063
Political threat	−0.230 ***	−0.200 ***	−0.114 ***	−0.002
Health threat	0.137 ***	0.021	0.033	0.109 **
Economic threat	−0.114 ***	−0.083 *	−0.159 ***	−0.133 ***
Explained variance (R^2^)	21%	15%	17%	0.02%

* *p* < 0.05, ** *p* < 0.01, *** *p* < 0.001; Factor 1 = belief in the government and prime minister; Factor 2 = belief in civil society; Factor 3 = patriotism; Factor 4 = belief in national bodies.

## Data Availability

The data collected in the longitudinal study is stored and kept by the authors.

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
