# Peer review of "Fluctuations in National Resilience during the COVID-19 Pandemic"

_ijerph, 2021, doi:10.3390/ijerph18083876_

Round 1

Reviewer 1 Report

Thank you for the opportunity to review the paper entitled “Fluctuations in National resilience along the COVID-19 pandemic.”

In this paper, the authors examined change in national resilience over time and factors related to national resilience under the COVID-19 pandemic situation.

The theme of the article is significant.

However, there are several points that I can recommend the article to be published in Int. J. Environ. Res. Public Health.

I hope these comments would be tips to improve the article.

  1. Introduction

“National resilience” is a keyword in this study. But the explanation of this concept does not enough. Please clearly elaborate it with more citations, including the theoretical components of the concept, who and how assess the NR, as well as how this concept has been developed, and recent findings of the significant studies about NR.

  1. Introduction

The authors mentioned “community resilience” and “national resilience” (L.54). Please describe the difference between these two concepts.

  1. Introduction

The methods in this study is included in the Introduction section (L.64-74).

  1. Introduction

Please describe the finding of the previous two-wave study by the same authors [9]. Besides, please add the explanation why the current three-wave study is newly needed. Also, while both these two studies with internet survey are very similar, the sample sizes are different. Thus please explain the differences between these studies.

  1. Introduction

I could not understand the meaning of the sentence “... the components of NR would be predicted by perceived levels of the health threat.” Please explain it more theoretically and logically.

  1. Methods

The demographic description of the participants in this study should be shown in not Methods section but Results section.

  1. Methods

The results of the factor analyses should also be shown in Results section. The authors may add how to conduct the factor analyses in Methods section.

  1. Discussion

Please mention about the implication for the practice increase how to increase national resilience based on the finding in this study.

Author Response

We would like to thank the reviewer for the wise and thoughtful feedback. We revised the manuscript accordingly and believe that it strengthened the article.

Reviewer 1

In this paper, the authors examined change in national resilience over time and factors related to national resilience under the COVID-19 pandemic situation.

The theme of the article is significant. However, there are several points that I can recommend the article to be published in Int. J. Environ. Res. Public Health.

I hope these comments would be tips to improve the article.

  1. Introduction

“National resilience” is a keyword in this study. But the explanation of this concept does not enough. Please clearly elaborate it with more citations, including the theoretical components of the concept, who and how to assess the NR, as well as how this concept has been developed, and recent findings of the significant studies about NR.

Response: We have added a paragraph regarding the explanation of individual, community, and national resilience and the different focuses of each of them (p. 2, lines 43-60). We also added references regarding national resilience (see references # 5-8 and 13).

  1. Introduction

The authors mentioned, “community resilience” and “national resilience” (L.54). Please describe the difference between these two concepts.

Response: We have added a short explanation regarding the difference between national and community resilience (p. 2, lines 49-59)

  1. Introduction

The methods in this study are included in the Introduction section (L.64-74).

Response: According to the reviewer's comment, we have revised these lines to avoid referring to the methods.

  1. Introduction

Please describe the finding of the previous two-wave study by the same authors [9]. Besides, please add an explanation of why the current three-wave study is newly needed. Also, while both these two studies with internet surveys are very similar, the sample sizes are different. Thus, please explain the differences between these studies.

Response: We have added an explanation regarding the difference between the previous study (T2 measurement) and the current study (T3 measurement) and how it can teach us about the relative importance of each of the three threats (health, economic and political) regarding national resilience (p. 2-3, lines 93-109). The lower level of response in the third measurement is derived from an expected reduction in response levels; since the same cohort of respondents was approached in all three measurements, as is common in most longitudinal studies, a decrease of 17% was noted between the second and third measurement. Nonetheless, the overall size of the sample was much higher than the representative sample that was calculated as required through OpenEpi (N=384),

5. Introduction

I could not understand the meaning of the sentence “... the components of NR would be predicted by perceived levels of the health threat.” Please explain it more theoretically and logically.

Response: According to reviewer comments we have changed the request paragraph (p. 3, lines 102-108).

 6. Methods

The demographic description of the participants in this study should be shown not in the Methods section but in the Results section.

Response: the demographic description of the participants was moved to the results section; see page 5, line 185.

7. Methods

The results of the factor analyses should also be shown in the Results section. The authors may add how to conduct the factor analyses in the Methods section.

Response: The results of the Factor analysis were moved to the results section. See page 6, lines 187-200.

8. Discussion

Please mention the implication for the practice increase how to increase national resilience based on the finding in this study.

Response: We have added at the end of the conclusions part regarding practical implications for the increase of NR during and after the crisis. See page 11, lines 345-351.

Reviewer 2 Report

The  study measured the national resilience in three different time frames  along the COVID-19 pandemic in Israel (N=804), aiming to investigate t the direction and  changes during the crisis, and the predictors of NR. The study is very interesting. The sample is impressive considering the three detection times. 

In my opinion, however, there are aspects that can be improved.

  1. The NR construct should be better explained. I would start from the construct of individual resilience to arrive at the national team.
  2. The question of trust in institutions is very interesting. I expect it to be properly introduced in the first part of the manuscript.
  3. Methodology: factor 4 "belief in the public Israeli bodies" presented a  Cronbach Alpha =.60, please insert this as a limit.
  4. Conclusions are too concise. The application dimension is missing. Instead, I believe that these results have an importance in the public policies of states also for the European countries.

Author Response

We would like to thank the reviewer for the wise and thoughtful feedback. We revised the manuscript accordingly and believe that it strengthened the article.

Comments and Suggestions for Authors

The study measured the national resilience in three different time frames along the COVID-19 pandemic in Israel (N=804), aiming to investigate t the direction and changes during the crisis, and the predictors of NR. The study is very interesting. The sample is impressive considering the three detection times. 

In my opinion, however, some aspects can be improved.

  1. The NR construct should be better explained. I would start from the construct of individual resilience to arrive at the national team.

Response: According to the reviewer suggested, we added an explanation regarding the construct of individual, community, and national resilience (p. 2, line 46-60)

2. The question of trust in institutions is very interesting. I expect it to be properly introduced in the first part of the manuscript.

Response: We have added introducing the question of trust in national and other institutions in the introduction (p. 2, line 67-69) in addition to a more extensive explanation at the result part, dealing with factor analysis of NR scale (p. 6, line 200).

3. Methodology: factor 4 "belief in the public Israeli bodies" presented a  Cronbach Alpha =.60, please insert this as a limit.

Response: Considering that factor 4 factor includes only 3 items, the Alpha Cronbach of .60, is considered moderate, but acceptable (see Nunnally & Bernstein, 1994; Pallant, 2001). Nonetheless, we have added this as a limitation. See page 11, lines 323-324.

4. Conclusions are too concise. The application dimension is missing. Instead, I believe that these results have an importance in the public policies of states also for the European countries.

Response: We have added the applicability of the findings to the conclusions section. See page 11, lines 345-351.

Round 2

Reviewer 1 Report

Thank you for the current opportunity to review the revised manuscript.

I confirmed that points that I suggested were revised appropriately.